# Production of Oil and Phenolic-Rich Extracts from *Mauritia flexuosa* L.f. Using Sequential Supercritical and Conventional Solvent Extraction: Experimental and Economic Evaluation [†]

Ivan Best [1,2,*], Zaina Cartagena-Gonzales [1], Oscar Arana-Copa [1], Luis Olivera-Montenegro [1] and Giovani Zabot [3]

[1] Grupo de Ciencia, Tecnología e Innovación en Alimentos, Universidad San Ignacio de Loyola, Lima 15024, Peru; zaina.cartagena@usil.pe (Z.C.-G.); oscar.arana@usil.pe (O.A.-C.); lolivera@usil.edu.pe (L.O.-M.)

[2] Instituto de Ciencias de los Alimentos y Nutrición (ICAN-USIL), Universidad San Ignacio de Loyola, Lima 15024, Peru

[3] Laboratory of Agroindustrial Processes Engineering (LAPE), Federal University of Santa Maria (UFSM), Cachoeira do Sul 97105-900, Brazil; giovani.zabot@ufsm.br

\* Correspondence: ibest@usil.edu.pe; Tel.: +51-1-3171000

† This paper is an extended version of paper published in the international conference: The 2nd International Electronic Conference on Foods 2021-Future Foods and Food Technologies for a Sustainable World, sciforum-048831, https://sciforum.net/paper/view/10988.

**Abstract:** *Mauritia flexuosa* L.f. is a palm from the Amazon. Pulp and oil are extracted from its fruits, with a high content of bioactive compounds. This study presents the economic evaluation of two extraction processes: (a) Conventional solvent extraction (CSE) with 80% ethanol for the recovery of phenolic-rich extracts; and (b) Supercritical fluid extraction (SFE) followed by CSE to obtain oil and phenolic-rich extracts. The objective of this study was to compare the feasibility of both extraction processes. The economic evaluation and the sensitivity study were evaluated using the SuperPro Designer 9.0® software at an extraction volume of 2000 L. Similar global extraction yields were obtained for both processes; however, 8.4 and 2.4 times more total polyphenol and flavonoid content were extracted, respectively, using SFE+CSE. Cost of manufacturing (COM) was higher in SFE+CSE compared to CSE, USD 193.38/kg and USD 126.47/kg, respectively; however, in the first process, two by-products were obtained. The sensitivity study showed that the cost of the raw material was the factor that had the highest impact on COM in both extraction processes. SFE+CSE was the most economically viable process for obtaining bioactive compounds on an industrial scale from *M. flexuosa* L.f.

**Keywords:** *Mauritia flexuosa* L.f.; conventional solvent extraction; supercritical fluid extraction; phenolic compounds; economic analysis

## 1. Introduction

*Mauritia flexuosa* L.f. is a palm from the South American Amazon and it is distributed in Peru, Bolivia, Brazil, Colombia, Ecuador, Venezuela and Guyana [1]. The fruit of *M. flexuosa* is considered a functional food due to its high content of phenolic compounds, carotenoids, essential fatty acids, vitamin E (tocopherols) and dietary fiber [2–4]. Moreover, from the pulp, 20–30% (wt.) of oil can be extracted [5], which contains 89.81% and 10.19% of unsaturated and saturated fatty acids, respectively, as well as a high content of β-carotene (911.4 mg/kg) and tocopherol (800 mg/kg) [6,7]. Oleic acid, a monounsaturated fatty acid, is the most abundant (89.81%) compound in the oil, followed by palmitic acid and linoleic acid [7,8].

The phenolic compounds extracted from *M. flexuosa* have anti-inflammatory, antioxidant, and antimicrobial properties [1–4,9], important for the prevention of chronic or

non-chronic diseases, which have great potential in the food, pharmaceutical and cosmetic industries for the development of new products as colorants, flavorings, additives, antimicrobials and antioxidants [10]. On the other hand, *M. flexuosa* oil could be used in the cosmetic industry for the treatment of skin and hair due to its high content of carotenoids and vitamin C, which are related to its high antioxidant activity [11]. Interestingly, *M. flexuosa* oil also has antimicrobial activity [8,12] with potential application in the food industry.

Currently, the COVID-19 pandemic has increased the size of the global nutraceutical market, corresponding to $ 417.66 billion in 2020, which is projected to grow at a compound annual growth rate (CAGR) of 8.9% from 2020 to 2028. Within this market, the segments that have shown the highest growth are dietary supplements and functional foods [13]. *M. flexuosa* oil and its bioactive compounds also have a high nutraceutical value. It is estimated that there are approximately 4000 vegetable species from which oil can be extracted [14]. There is not much information regarding the market value of oil extracted from tropical fruits such as *M. flexuosa*. However, omega-6 and omega-9 fatty acids obtained from the hydrolysis *of M. flexuosa* oil represent important products with high added value in the cosmetic, food and pharmaceutical industries [7].

For the recovery of phenolic-rich extracts from the pulp of *M. flexuosa* and fat-soluble compounds from the oil of this fruit, solid–liquid extraction [2–4] and supercritical adsorption in columns packed with γ-alumina [5] were used, respectively. In some of these processes, large volumes of petroleum-derived solvents are required, as well as a long extraction time, which could reduce the quality of the bioactive compounds obtained [15]. One strategy for reducing costs without affecting the quality of these products is the intensification of processes that allow efficient use of energy and capital, improving the techno-economic parameters [16].

Using the concept of biorefinery and through the sequential integration of green extraction processes, the yield and recovery of bioactive compounds can be increased on an industrial scale, to be used as functional foods, as well as in the food, pharmaceutical and cosmetic industry [17]. However, before starting up an industrial-scale biorefinery, it is important to know all the production costs that would be associated with its implementation.

A previous study based on the fruits of *M. flexuosa* shows that the sequential use of a supercritical and a conventional solvent extraction, compared to conventional solvent extraction alone, makes it possible to obtain two by-products with high nutraceutical and commercial value: oil and phenolic-rich extracts [18]. Therefore, the objective of this study was to carry out an economic evaluation and sensitivity study of the by-products generated through the two extraction processes: a single-stage process with conventional solvent extraction and a two-stage sequential process using supercritical and conventional solvent extraction, at an extraction volume of 2000 L.

## 2. Materials and Methods

### 2.1. Sample Preparation

Fruits of *M. flexuosa* of the "Shambo" morphotype, acquired in October 2018 in the "Veinte de Enero" Community of the Marañon River, Iquitos Region, Peru (latitude: 4°39′19.5″ S, longitude: 73°49′27.9″ W), were used in this study. The fruits were selected from their sanity and ripening stage, and washed in water containing 25 ppm of sodium hypochlorite. Then, the pulp was obtained, which was lyophilized for subsequent assays as previously described [3].

### 2.2. Single-Stage Process and Two-Stage Sequential Process

A supercritical $CO_2$ extraction equipment (Top Industrie, Vaux-le-Pénil, France) was used to obtain oil from *M. flexuosa* on a laboratory scale. The optimized conditions using SFE to maximize oil extraction were: pressure, $2 \times 10^7$ Pa; extraction temperature, 42 °C and $CO_2$ flow rate, 42 g $CO_2$/min. For each extraction, the 50 mL extraction vessel was filled with approximately 50 g of lyophilized *M. flexuosa* pulp.

Phenol-rich extracts under previously optimized conditions were obtained from SFE defatted pulp or freeze-dried pulp of *M. flexuosa* on a laboratory scale, as previously described [3]. During this study, two extraction processes were evaluated: (a) Single-stage process by conventional solvent extraction (CSE) for obtaining phenolic-rich extracts and (b) Two-stage sequential process using supercritical and conventional solvent extraction (SFE+CSE) for the recovery of oil and phenolic-rich extracts.

The global extraction yields (GEY) for both extraction processes were calculated as the ratio between the total mass of extract and the mass of raw material loaded in the extractor on a dry weight (dw) basis [19].

### 2.3. Total Phenolic and Flavonoid Content

Total phenolics were extracted using a modified Folin–Ciocalteau method as described in Best et al. [3]. Briefly, 750 µL of 0.2 N Folin–Ciocalteau reagent was added to 100 µL of extract and allowed to react for 5 min. Then, 750 µL of a 7.5% sodium carbonate solution was added to the mixture and it was incubated in a water bath at 40 °C for 30 min. After this time, the absorbance at 725 nm was recorded. Total phenolics were expressed in µg of gallic acid equivalents per g of sample (µg GAE/g).

Total flavonoids were measured using the aluminum chloride colorimetric method as previously described [3]. First, 75 µL of a 5% $NaNO_2$ solution was added to 100 µL of extract and kept for 5 min at 25 °C. Then, 150 µL of a 10% $AlCl_3.6H_2O$ solution was added to the mixture and it was incubated for 5 min. Subsequently, 500 µL of 1 M NaOH was added and it was left to react for 15 min at room temperature. After this time, the absorbance at 510 nm was read. Total flavonoids were expressed as µg of catechin equivalents per g of sample (µg CE/g).

### 2.4. Process Simulation Model

SuperPro Designer 9.0® software was used to perform the simulations of the CSE and SFE+CSE. Direct costs (buildings, yard improvement, electrical facilities, insulation, instrumentation, installation, etc.) and indirect costs (administration rates, engineering, and construction, insurance, human resources for administration, cleaning services, etc.) were also estimated by the simulator, and both are considered in the economic evaluation.

The input parameters and simulation conditions for the single-stage process (CSE) and sequential two-stage process (SFE+CSE) are shown in Tables 1 and 2.

**Table 1.** Experimental data used to simulate the single-stage process by conventional solvent extraction (CSE).

| Parameter | Value |
|---|---|
| Lyophilized pulp—1st step | |
| Lyophilized yield | 36.17 g/100 g whole fruit |
| Lyophilization temperature | −50 °C |
| Time | 3–4 h |
| Pressure | ≥50 Pa |
| Conventional solvent extraction (80% ethanol)—2nd step | |
| Extraction yield | 87.3 g ground and lyophilized pulp/100 g lyophilized pulp |
| Temperature | 30 °C |
| Time | 1 h |
| S/F | 10 m ethanol/1 g lyophilized pulp |
| Lyophilized extract—3rd step | |
| Lyophilized yield | 85.44 g lyophilized extract/100 g lyophilized pulp |
| Lyophilization temperature | −50 °C |
| Time | 3–4 h |
| Pressure | ≥50 Pa |

S/F: mass ratio of solvent to feed.

**Table 2.** Experimental data used to simulate the sequential two-stage process using supercritical and conventional solvent extraction (SFE+CSE).

| Parameter | Value |
|---|---|
| Lyophilized pulp—1st step | |
| Lyophilized yield | 36.17 g/100 g whole fruit |
| Lyophilization temperature | $-50\,^{\circ}C$ |
| Time | 3–4 h |
| Pressure | $\geq 50$ Pa |
| Oil extract—2nd step | |
| Extraction yield | 38.85 g oil/100 g lyophilized pulp |
| Temperature | $80\,^{\circ}C$ |
| Extraction time | 1 h |
| Pressure | $2 \times 10^7$ Pa |
| $CO_2$ flow rate | 42 g $CO_2$/min |
| Lyophilized extract–3rd step | |
| Lyophilized yield | 47.418 g lyophilized extract/100 g lyophilized pulp |
| Lyophilization temperature | $-50\,^{\circ}C$ |
| Extraction time | 3–4 h |
| Pressure | $\geq 50$ Pa |

In Figure 1, the flowsheets of the CSE and SFE+CSE are shown. During the scale-up process, it was observed that the yield obtained on an industrial scale can increase the extraction yield compared to the laboratory scale, under the same processing conditions for each technology (pressure, temperature, extraction time, density) [20].

Both extraction processes include a wash tank (P-0/WSH-101), a disinfection tank (P-02/WSH-103), a rinse tank (P-03/WSH-104), a maturing kettle (P-04/V-104), a bleaching kettle (P-05/V-105), a pulping machine (P-06/SR-101), a packaging machine (P-07/SL-101), a freezer (P-08/FT-101), a lyophilizer (P-09/V-103), a plate filter (P-10/GR-101), and a sieving machine (P-11/VSCR-101). In the CSE, the lyophilizate was placed in an extraction tank (P-12/MSX-101), then it was centrifuged (P-13/DS-101, filtered by plates (P-14/NFD-10), evaporated (P-15/EV-101), packed (P-16/SL-102), frozen (P-17/FT-102), and lyophilized (P-18/V-101) until obtaining phenolic-rich extracts.

On the other hand, in the SFE+CSE, the lyophilizate was placed in the supercritical $CO_2$ equipment (P-12/V-102, P-13/V-106, P-14/G-101, P-15/MX-101, P-16/EC-101) for oil separation, then the aqueous phase was pumped (P-17/PM-101) into the extraction tank (P-18/MSX-101), then it was centrifuged (P-19/DS-101), plate filtered (P-20/NFD-101), evaporated (P-21/EV-101), packed (P-22/SL-102), frozen (P-23/FT-102), and lyophilized (P-24/V-101) to obtain the second by-product of this process: phenolic-rich extracts.

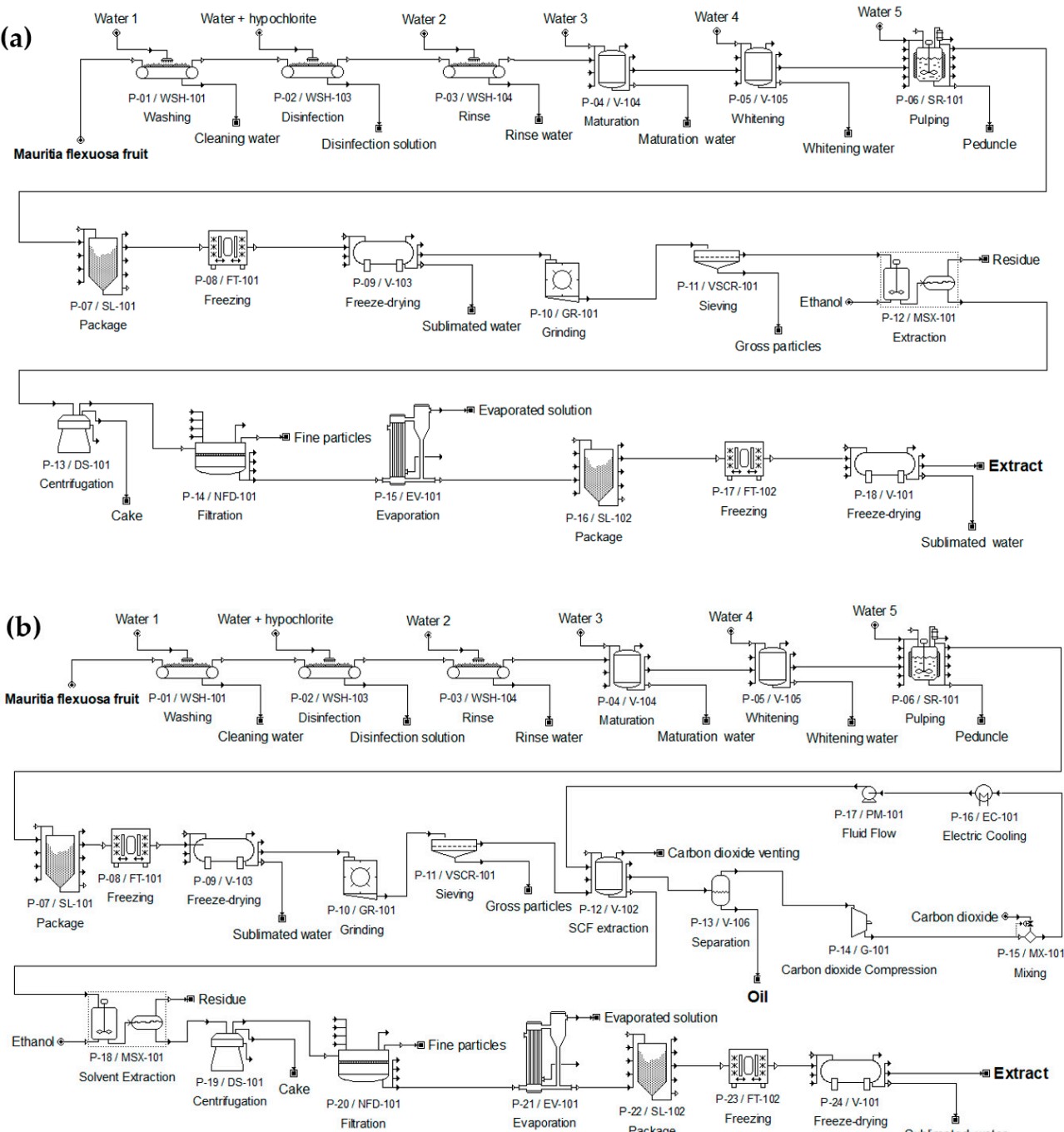

**Figure 1.** Flowsheets of the (**a**) single-stage process by conventional solvent extraction (CSE) and (**b**) sequential two-stage process using supercritical and conventional solvent extraction (SFE+CSE), designed using SuperPro Designer 9.0® software. Source: Ref. [18], reproduced with permission from Best et al., The 2nd International Electronic Conference on Foods 2021-Future Foods and Food Technologies for a Sustainable World, sciforum-048831; published by MDPI, 2021.

*2.5. Economic Evaluation*

The cost of the extraction plants for the CSE and SFE+CSE was calculated using past quotes from vendors and previous reports [21]. In some cases, the quotes and detailed specifications of the equipment were of different capacities than those required [22]. Equation

(1) was used to obtain the cost of each large-scale equipment based on the quote obtained for small-scale equipment.

$$C_1 = C_2 \left( \frac{Q_1}{Q_2} \right)^n \tag{1}$$

where $C_1$ is the cost of the equipment with capacity $Q_1$, $C_2$ is the known base cost for equipment with capacity $Q_2$, and n is a constant depending on the equipment type. The values of n were collected from the literature [23–26]. The cost of the supercritical fluid equipment was calculated according to [27]. Unit base cost and n values used for the extraction plants for the CSE and SFE+CSE are shown in Table 3.

**Table 3.** Base costs for each equipment composing the extraction plants.

| Equipment | N [a] | Unit Base Cost (USD) | CSE Plant | | SFE+CSE Plant | |
|---|---|---|---|---|---|---|
| | | | Number of Equipment | Total Base Cost (USD) | Number of Equipment | Total Base Cost (USD) |
| Sorting machine [b] | 0.89 | 3900.00 | 1 | 1,824,167.05 | 1 | 1,824,167.05 |
| Immersion washer (washed) [b] | 0.53 | 3937.74 | 2 | 306,391.73 | 2 | 306,391.73 |
| Immersion washer (rinsed) [b] | 0.53 | 3937.74 | 2 | 306,391.73 | 2 | 306,391.73 |
| Rinse tank [b] | 0.53 | 4000.00 | 4 | 622,472.23 | 4 | 622,472.23 |
| Water boiler (maturation) [b] | 0.59 | 2500.00 | 4 | 588,843.66 | 4 | 588,843.66 |
| Water boiler (bleached) [b] | 0.59 | 2500.00 | 4 | 588,843.66 | 4 | 588,843.66 |
| Automatic pulper [b] | 0.60 | 1895.73 | 5 | 598,062.38 | 5 | 598,062.38 |
| Automatic packaging machine [b] | 0.60 | 1650.00 | 1 | 104,107.96 | 1 | 104,107.96 |
| Freezing tunnel [b] | 0.63 | 2500.00 | 1 | 194,061.78 | 1 | 194,061.78 |
| Lyophilizer [b] | 0.65 | 20,000.00 | 1 | 1,782,501.88 | 1 | 1,782,501.88 |
| Roller mill [b] | 0.91 | 5700.00 | 1 | 3,061,081.24 | 1 | 3,061,081.24 |
| Industrial sieve [b] | 0.91 | 1700.00 | 1 | 912,954.05 | 1 | 912,954.05 |
| Extraction tank [b] | 0.82 | 2000.00 | 1 | 576,806.30 | | |
| | | 1500.00 | | | 1 | 432,604.73 |
| Supercritical $CO_2$ equipment [c] | 0.60 | 2,520,106.41 | - | - | 1 | 159,007,964.56 |
| Centrifuge [b] | 0.71 | 7000.00 | 7 | 5,665,644.11 | 4 | 3,237,510.92 |
| Plate filter [b] | 0.66 | 1500.00 | 1 | 143,248.89 | - | - |
| | | 700.00 | - | - | 1 | 66,849.48 |
| Evaporator [b] | 0.59 | 10,000.00 | 6 | 706,612.39 | 3 | 353,306.19 |
| Lyophilizer [b] | 0.65 | 15,000.00 | 1 | 1,336,876.41 | - | - |
| | | 10,000.00 | - | - | 1 | 891,250.94 |
| Conveyor belts [b] | 0.89 | 769.00 | 24 | 8,632,519.77 | 24 | 8,632,519.77 |
| Centrifugal pump [b] | 0.55 | 900.00 | 5 | 201,007.62 | 5 | 201,007.62 |
| Total | - | - | - | 28,152,594.81 | - | 183,712,893.54 |

[a] n constant depending on equipment type based on references [23–26]. [b] Direct quotation. [c] Calculated based on [27].

For both extraction processes, the scale-up was carried out for a vessel with a volume of 2000 L. To perform the simulations, process operation of three daily shifts for 330 days per year was considered, corresponding an annual operation for 7920 h. For each batch, two tons of *M. flexuosa* were processed in both CSE and SFE+CSE.

The cost of raw material (*M. flexuosa*) was quoted as USD 15.63/kg (direct quotation of wholesale market, Lima, Peru in 2021). The commercialization of phenolic-rich extracts obtained by the CSE and SFE+CSE was estimated at USD 100.00/kg and USD 180.00/kg, respectively. The commercialization of oil was estimated at USD 314.47/L. The other input information is shown in Table 4.

**Table 4.** Input economic parameters used in SuperPro Designer 9.0[®] software.

| Pameter | Value |
| --- | --- |
| Fixed Capital Investment (FCI) | |
| CSE plant [a] | USD 28,152,594.00 |
| SFE+CSE plant [a] | USD 183,712,894,00 |
| Depreciation rate [b] | 10%/year |
| Maintenance rate [b] | 6%/year |
| Project lifetime | 25 years |
| Inflation | 4%/year |
| Low NPV interest | 7% |
| Depreciation period | 25 years |
| Loan period for equipment | 12 years |
| Loan interest for equipment | 7%/year |
| Loan | 100% |
| Cost of operational labor (COL) | |
| Wage (with administration and benefits) [c] | USD 4.91/h |
| Number of workers per shift | 8 |
| Operational time | 7920 h/year |
| Cost of Raw Material (CRM) | |
| *Mauritia flexuosa* L.f. [a] | 15.63 USD/kg |
| Industrial $CO_2$ [a] | 0.033 USD/kg |
| Ethanol 80% [a] | 0.53 USD/kg |
| Cost of utilities (COU) | |
| Electricity | 0.1183 USD/kW.h |
| Steam | 12 USD/ton |
| Water | 1.63 USD/ton |

[a] Based on local quotations. [b] Calculated based on reference [23]. [c] Based on reference [28].

Experimental data obtained at fixed operating conditions were used as input for the model. The cost of manufacturing (COM) for the production of phenolic-rich extracts by the CSE, as well as the production of oil and phenolic-rich extracts by SFE+CSE, was determined as the sum of three main components: direct costs, fixed costs, and general expenses. COM was estimated according to a methodology proposed elsewhere [26], in which the three main components are estimated in terms of four major costs: fixed capital investment (FCI), cost of raw material (CRM), cost of operational labor (COL), and cost of utilities (CUT). The FCI is related to expenses involved in the implementation of the production plant. CRM considers the cost of the raw material, including the costs of the extraction solvents. COL is related to the number of operators required to perform all stages of extraction. CUT considers electricity requirements, steam, and treated water for the process.

*2.6. Sensitivity Study*

The simulation was carried out considering an industrial scale at an extraction volume of 2000 L. The value of COM was simulated in CSE and SFE+CSE, considering six different scenarios: (1) Normal or real value of COM; (2) Plant at 50% the cost; (3) *M. flexuosa* at 50% the cost; (4) Ethanol 50% recycled; (5) Extract lyophilized 50% more expensive; and (6) Merging scenarios 2–5.

In addition to COM, to carry out the sensitivity study, the gross margin (GM), return over the investment (ROI), payback time (PBT), internal rate of return (IRR), and net present value (NPV) at 7% interest were also simulated considering the above-mentioned selling prices of oil and phenolic-rich extracts.

*2.7. Statistical Analysis*

The results were expressed as mean $\pm$ standard deviation (SD), and analyzed using Statistical Package for the Social Sciences (SPSS v26.0, IBM, Chicago, IL, USA). Differences

between groups were evaluated using the Mann–Whitney U test, at a significance level of $p < 0.05$.

## 3. Results and Discussion

### 3.1. Experimental Results

As shown in Table 5, in the CSE, the global extraction yield was 13.84 g extract/100 g *M. flexuosa* pulp (dry basis), while in the SFE+CSE, the global extraction yield was 44.5 g oil/100 g *M. flexuosa* pulp (dry basis) and 13.84 g extract/100 g *M. flexuosa* pulp (dry basis). These results are in agreement with previous studies showing an extraction yield of 8.04% for phenolic-rich extracts obtained from the pulp of *M. flexuosa* defatted by Sohxlet [29] and an oil extraction yield between 23.5 to 41.1 g oil/100 g *M. flexuosa* using SFE-$CO_2$ [30]. A study carried out in Brazil showed that the annual productivity of pulp and oil from *M. flexuosa* was $0.79 \pm 0.23$ t/ha and $57.5 \pm 17.0$ kg/ha, respectively [31]. Currently, the average cost of *M. flexuosa* oil in Peru is USD 314.47 L; however, to increase its productivity, due to its high antioxidant potential, it is necessary to extract it on an industrial scale.

**Table 5.** Global extraction yields, total phenolics, and total flavonoids in the extracts obtained by the single-stage process by conventional solvent extraction (CSE) and the sequential two-stage process using supercritical and conventional solvent extraction (SFE+CSE).

| | CSE | SFE+CSE | Reference |
|---|---|---|---|
| Global extraction yield (%) | 13.84% (extract) | 44.85% (oil) 13.8% (extract) | - |
| Total phenolics (µg GAE/g extract) | $3423.94 \pm 24.93$ | $28800.95 \pm 1180.37$ * | Best et al. [3] |
| Total flavonoids (µg CE/g extract) | $165.34 \pm 4.11$ | $390.82 \pm 21.11$ * | Best et al. [3] |

CSE: Conventional solvent extraction; SFE: Supercritical fluid extraction; GAE: Gallic acid equivalents; CE: Catechin equivalent. * Mann–Whitney's U test, $p < 0.05$.

In the SFE+CSE, the total content of polyphenols and flavonoids were 8.4- and 2.4-fold higher, respectively, compared to the CSE ($p < 0.01$). The levels of total polyphenols obtained by the CSE were similar to those reported by previous studies in methanol extracts from *M. flexuosa* pulp [9,32]. However, these levels were significantly lower compared to those found from the pulp defatted by the SFE+CSE. This last method allows to concentrate the content of phenolic compounds, and therefore to increase the activity and the market value of this phenolic-rich extracts from *M. flexuosa*. Regarding the total flavonoid levels, the content obtained by both extraction methods in the present study was lower than that reported by previous studies [9,32].

### 3.2. Economic Evaluation of the Extraction Processes

The total investment for CSE and SFE+CSE was USD 28,152,594.00 and USD 183,712,894.00, respectively (Table 4). These differences in the cost of the total investment are due to the use of a supercritical fluid equipment in the SFE+CSE, which makes it possible to obtain two by-products: oil and phenolic-rich extracts. For the CSE, the productivity was 731.1 tons extract/year, while for the SFE+CSE, the productivity was 335.9 tons oil/year and 405.8 tons extract/year (Figures 2 and 3).

In the sensitivity study, for both extraction processes, no variation was observed in the productivity of oil and/or phenolic-rich extracts among all the evaluated scenarios. The differences on the COM and productivities of the by-products obtained by both extraction processes were related to the input data for the simulation, extraction yields, and total investment cost of each process.

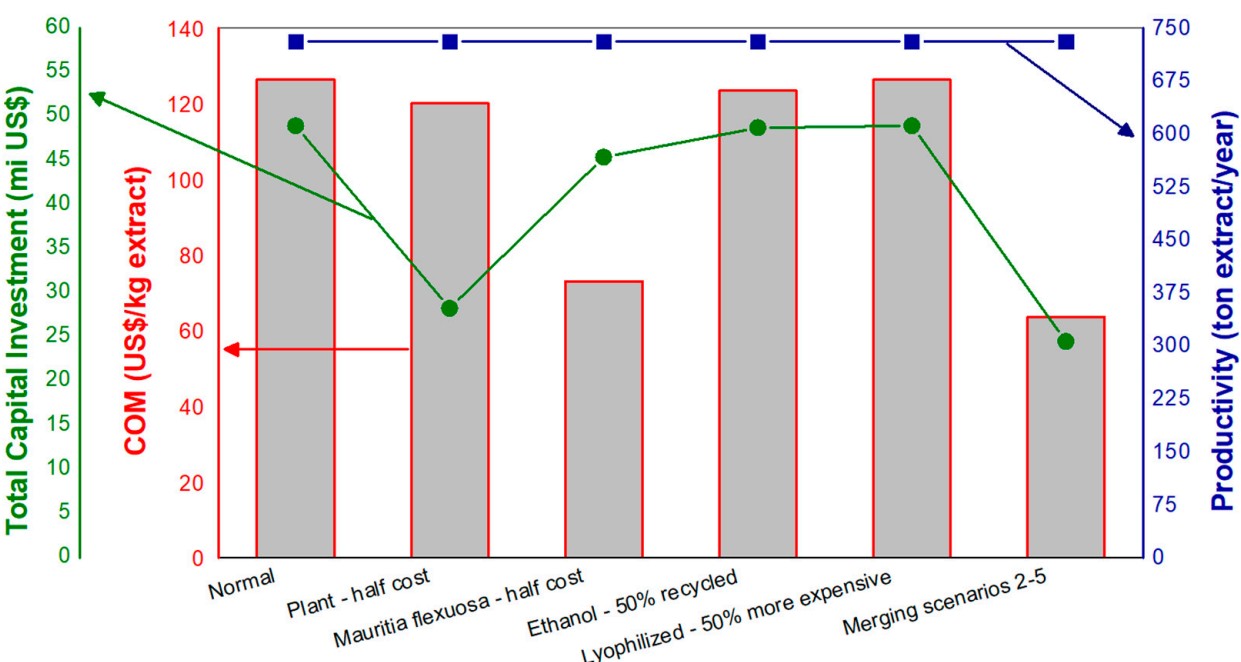

**Figure 2.** COM, productivity, and total capital investment to produce phenolic-rich extracts by the single-stage process using conventional solvent extraction (CSE).

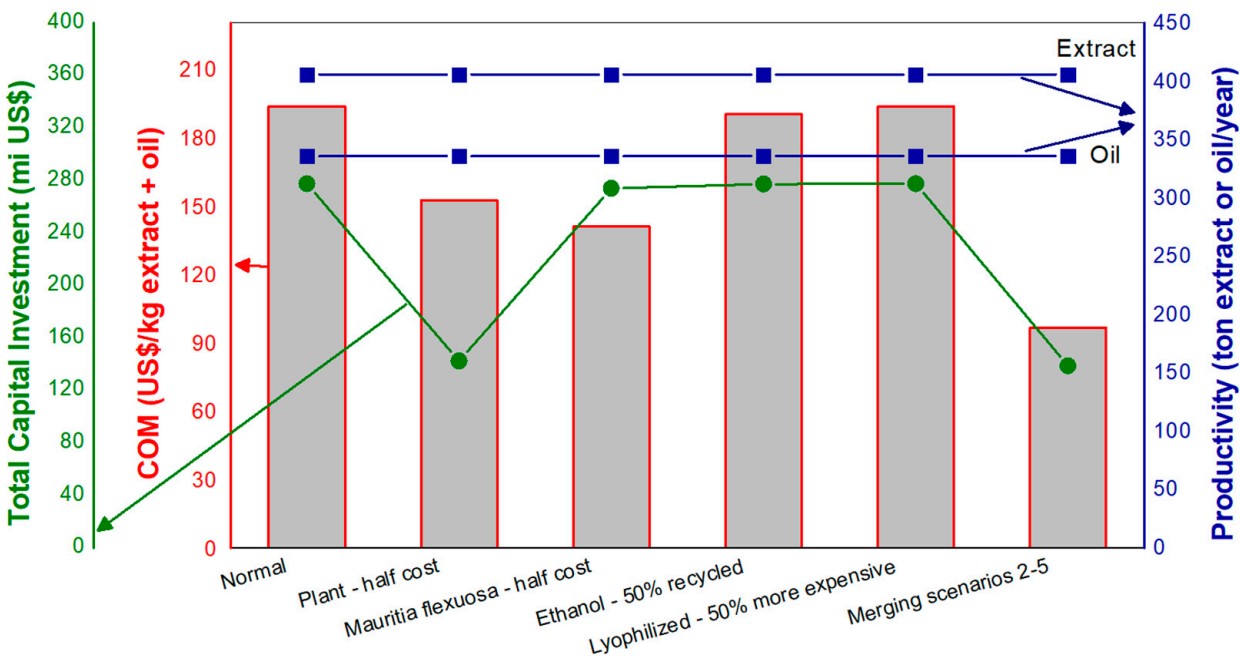

**Figure 3.** COM, productivity, and total capital investment to produce oil and phenolic-rich extracts by the sequential two-stage process using supercritical and conventional solvent extraction (SFE+CSE).

Figure 4 shows the contribution of the main cost factors (CRM, COL, FCI and CUT) on COM for each extraction process. For CSE and SFE+CSE, the CRM and FCI were the components that presented the highest contribution to COM. CRM corresponded to 89.22% and 57.02% of the COM in the CSE and SFE+CSE, respectively; while the FCI represented 10.10% and 42.48% of the COM in the CSE and SFE+CSE, respectively. Other costs such as CUT and COL had a lesser influence on the COM, together representing 0.69%

and 0.51% in the CSE and SFE+CSE, respectively. As the production capacity of a plant increases, the CUT and COL increase and decrease, respectively [33]. In the present study, the CUT and COL values were not very significant, since different production capacities were not evaluated.

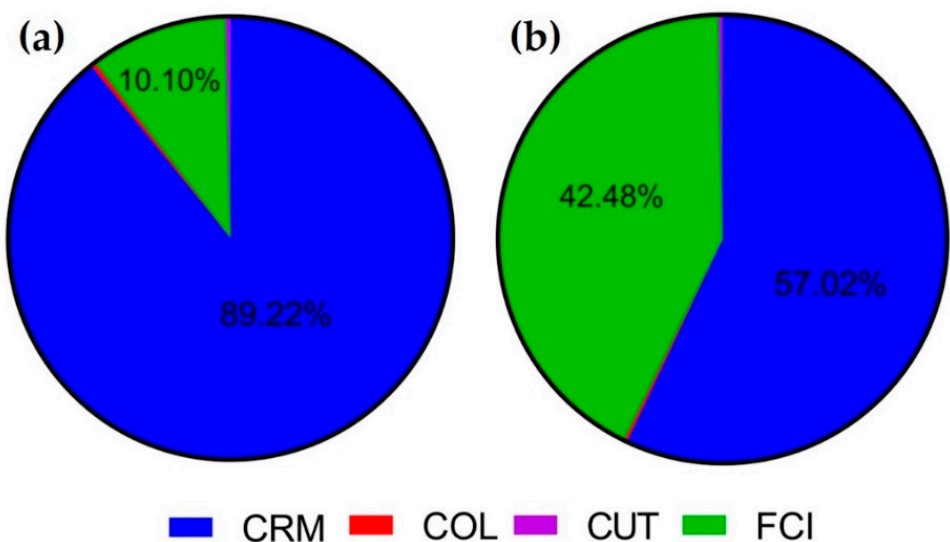

**Figure 4.** Contribution of each component (CRM, CUT, COL, and FCI) on the COM for bioactive compounds of *M. flexuosa* obtained by (**a**) Conventional solvent extraction (CSE) and (**b**) Supercritical fluid and conventional solvent extraction (SFE+CSE).

These results suggest that the acquisition cost of the raw material in both extraction processes exerted a strong influence on the COM. The cost of fresh *M. flexuosa* corresponded to 96.52% of the CRM, while the cost of supercritical $CO_2$ and ethanol represented only 3.48%, and fresh *M. flexuosa* obtained at a more affordable cost could notably decrease the COM [34]. Moreover, the FCI was the cost that had the second highest impact on the COM; however, an increase in the extraction time from 930 min to 1145 min in the CSE and SFE+CSE, respectively, decreased the contribution of the CRM and increased the impact of the FCI on the COM due to an increase in the operation time and use of supercritical $CO_2$ equipment, as described by Rosa et al. [35] for the extraction of clove bud oil and ginger oleoresin using supercritical fluid technology.

### 3.3. Sensitivity Study and Comparison between Extraction Methods

As shown in Table 6, when the CSE was used considering six different scenarios, the COM of one kg of phenolic-rich extracts ranged between USD 63.63 and USD 126.47. The main influence on the COM was the CRM, which has an impact of approximately 80–90%. This occurs because *M. flexuosa* is a biomass of great importance in the Peruvian market, as well as in the rest of South America. When it is possible to purchase this raw material at half cost, the parameters of return indicate the feasibility of the process. For example, the GM, ROI, and IRR were 26.94%, 33.46% and 42.42%, respectively. The PBT was 2.99 years with an NPV of USD 123,274,000.00. The best scenario for processing *M. flexuosa* by the CSE was achieved when merging scenarios 2–5, decreasing the value of COM by approximately two times.

**Table 6.** Project indices of the single-stage process by conventional solvent extraction (CSE).

| Scenario | Condition | COM (USD/kg) | GM (%) | ROI (%) | PBT (Year) | IRR (%) | NPV (USD) (at 7% Interest) |
|----------|-----------|--------------|--------|---------|------------|---------|----------------------------|
| 1 | Normal | 126.47 | NA | NA | NA | NA | NA |
| 2 | Plant—half cost | 120.09 | NA | NA | NA | NA | NA |
| 3 | *M. flexuosa*—half cost | 73.06 | 26.94 | 33.46 | 2.99 | 42.42 | 123,274,000.00 |
| 4 | Ethanol—50% recycled | 123.44 | NA | NA | NA | NA | NA |
| 5 | Lyophilized extract—50% more expensive | 126.47 | 15.68 | 27.97 | 3.57 | 34.30 | 103,938,000.00 |
| 6 | Merging scenarios 2–5 | 63.63 | 57.58 | 161.53 | 0.62 | 324.14 | 417,069,000.00 |

NA: Not applicable; COM: Cost of manufacturing; GM: Gross margin; ROI: Return on investment; PBT: Payback time; IRR: Internal rate of return after taxes; NPV: Net present value at 7%. Source: Ref. [18], reproduced with permission from Best et al., The 2nd International Electronic Conference on Foods 2021-Future Foods and Food Technologies for a Sustainable World, sciforum-048831; published by MDPI, 2021.

In the same trend, when the SFE+CSE was carried out, two by-products were obtained in each batch. COM of one kg of oil + phenolic-rich extracts ranged from USD 96.31 to USD 193.38. Overall, the values of COM were a bit higher in SFE+CSE than CSE because the SFE equipment increased the FCI cost significantly. This itemized cost contributed to approximately 50% of the total cost of these by-products. In this process, both CRM and FCI had a significant impact on the COM. Despite the values of COM being high, all scenarios for SFE+CSE presented positive returns on the initial capital and operational investment. The best scenario for processing *M. flexuosa* by SFE+CSE was achieved by merging scenarios 2–5, which decreased the COM by two times. In this scenario, the GM, ROI and IRR were 73.34%, 92.91% and 152.58%, respectively. The PBT was 1.08 years, with an NPV of USD 1,294,690,000.00 (Table 7).

**Table 7.** Project indices of the sequential two-stage process using supercritical and conventional solvent extraction (SFE+CSE).

| Scenario | Condition | COM (USD/kg) | GM (%) | ROI (%) | PBT (Year) | IRR (%) | NPV (USD) (at 7% Interest) |
|----------|-----------|--------------|--------|---------|------------|---------|----------------------------|
| 1 | Normal | 193.38 | 19.73 | 15.54 | 6.43 | 16.48 | 193,979,000.00 |
| 2 | Plant—half cost | 152.30 | 36.78 | 35.38 | 2.83 | 45.55 | 416,182,000.00 |
| 3 | *M. flexuosa*—half cost | 140.75 | 41.57 | 24.29 | 4.12 | 28.83 | 457,940,000.00 |
| 4 | Ethanol—50% recycled | 190.01 | 21.13 | 16.10 | 6.21 | 17.27 | 210,877,000.00 |
| 5 | Lyophilized extract—50% more expensive | 193.38 | 46.94 | 35.13 | 2.85 | 45.08 | 800,105,000.00 |
| 6 | Merging scenarios 2–5 | 96.31 | 73.34 | 92.91 | 1.08 | 152.58 | 1,294,690,000.00 |

COM: Cost of manufacturing; GM: Gross margin; ROI: Return on investment; PBT: Payback time; IRR: Internal rate of return after taxes; NPV: Net present value at 7%. Source: Ref. [18], reproduced with permission from Best et al., The 2nd International Electronic Conference on Foods 2021-Future Foods and Food Technologies for a Sustainable World, sciforum-048831; published by MDPI, 2021.

As shown in Tables 6 and 7, the COM calculated for the by-products of both extraction processes was lower than the sale price when scenarios 2 to 5 were merged, which suggests that both extraction processes are profitable under those conditions.

In 2021, the global market for plant extracts was USD 30.8 billion, forecast to reach USD 55.3 billion by 2026, with a CAGR of 6.0%. Within this market, phytomedicines, herbal extracts, essential oils and flavors are in greater demand. In the current scenario, due to the COVID-19 pandemic, the food and pharmaceutical industries have increased the consumption of plant extracts that enhance human immunity. However, the confinement and the increase in infections due to the emergence of new variants of SARS-CoV-2, has



limited the supply and transportation of raw materials from extracts of natural products, which, added to its high demand, will increase the cost of extracts from natural products during the global spread of this virus [36].

It is estimated that the COM of natural extracts ranges from USD 3.00/kg to USD 5000.00/kg [16]. In the present study, in both extraction processes, the calculated COM value was within the range for commercial extracts. Interestingly, in both cases, the COM value was lower than the commercial value of the extracts obtained by CSE and SFE+CSE (wholesale market, Lima, Peru in 2021). Raw materials can represent up to 80% of COM when supercritical fluid extraction is used [16]. According to Osorio-Tobón et al. [37], raw materials, despite their high variability in cost, are generally the components with the highest contribution to COM. In the present study, when scenarios 2 to 5 were merged, the COM was 1.5 higher in the SFE+CSE compared to the CSE; however, using the former process, two by-products were recovered. In the latter scenario, our COM for SFE+CSE was similar to the COM obtained for the extraction of essential oil and curcuminoid-rich extracts from *Curcuma longa* L using the supercritical fluid extraction, pressurized liquid extraction, and supercritical antisolvent processes [37]. Moreover, in the present study, the COM for phenolic-rich extracts using CSE was similar or even lower than the COM for extraction of phenolic compounds from the pulp of *Euterpe edulis* [38].

GM evaluates the short-term benefits of the extraction process [21], with a higher GM indicating that the project is more feasible, because this indicator represents the percentage of every dollar of a product sold that the company will retain as gross profit [21,39]. In general, for both extraction processes, the GM decreased as the COM increased, with a higher GM being observed when scenarios 2 to 5 were merged. For CSE, the GM was positive for a COM of USD 73.06. Similar results were found by Galviz-Quezada et al. [33], who obtained positive GM values for the extraction of phenolic compounds from iraca at selling prices higher than USD 100/kg. On the other hand, for SFE+CSE, positive GM values were obtained under all of the evaluated scenarios. However, in the most favorable scenario, the GM was significantly higher than that reported for the production of essential oil and curcuminoid-rich extracts from *C. longa* L. at a raw material cost of USD 7.27/kg and USD 1.59/kg [37].

Another important parameter for evaluating the performance of extraction processes is the ROI, where the higher the value, the more attractive the project [40]. However, for a project to be feasible, a minimum value of ROI between 10% to 15% is acceptable [21]. Similar to GM, in both extraction processes, ROI decreased as COM increased, reaching its highest value in both extraction processes when scenarios 2 to 5 were merged. When comparing both processes, it was observed that in the case of the CSE, the ROI was more significantly influenced by the costs of the raw materials and the extract, while for SFE+CSE, the costs of the plant and the extract had a higher impact on ROI, due to the higher level of investment in equipment necessary to carry out this process. For SFE+CSE, an ROI > 10% was achieved for a COM of USD 193.38, indicating that the maximum sale price for the production of oil and phenolic-rich extracts could be achieved using this process.

PBT is also an important parameter in the sensitivity study, making it possible to evaluate the time until the initial investment has been paid back. It is estimated that the shorter the PBT, the faster the initial investment will be recovered; however, this depends on the type of company and the investors [33]. For small and large plants, the PBT should be between 2 to 3 years and 7 to 10 years, respectively [21]. In the case of SFE+CSE, a time between 1.08 to 6.43 years was obtained when the COM was in the range from USD 96.31 to USD 193.38, indicating the feasibility of the process under all of the evaluated scenarios. When scenarios 2 to 5 were merged, SFE+CSE had a value of PBT that was 1.74 times higher than that of CSE, due to the higher level investment in equipment in SFE+CSE. A similar behavior was observed previously [36], with a PBT value in the range from 3.25 to 4.71 years in order to obtain two by-products. Similarly, PBT values ranging from 0.60 to 3.02 years were reported for obtaining oil from Sucupira Branca *(Pterodon emarginatus)* seeds by SFE with an extractor capacity of 30 L [41].

The IRR is another parameter used to evaluate the profitability of a project; similar to the ROI, the higher the IRR, the more desirable the project [33]. Internal rate of return (IRR) is an important parameter for assessing the profitability of a process, as it accounts for factors such as plant income, capital investment, and time value of money [42]. In general, for both extraction processes, the IRR increased when the COM decreased, reaching its highest value when scenarios 2 through 5 were merged. Similar to the ROI, in the CSE, it was observed that the costs of the raw materials and the extract had the highest impact on the COM, while in the SFE+CSE, the costs of the plant and the extract had a greater influence on the value of the COM. Therefore, in the most favorable scenario, the SFE+CSE showed an IRR 2.12 times lower than that of CSE. However, the IRR value in the SFE+CSE was similar to or even higher than that reported in other studies [37,43].

Finally, NPV assesses the present value of all future cash flows generated by a project, including the initial capital investment, making it possible to establish which projects are able to generate the most profit [33]. A project can be considered feasible if the NPV is positive after generally assuming an interest rate of 7% [37]. In the CSE, only scenarios 3, 5 and 6 present a positive NPV value, which would depend on the 50% reduction in the cost of raw material and the sale of the extract at prices that were 50% more expensive. For SFE+CSE, all of the evaluated scenarios presented a positive NPV value, which indicates that all of them are feasible. Similar to IRR, for both extraction processes, when scenarios 2 to 5 were merged, the NPV reached its highest value. In the latter scenario, SFE+CSE had an NPV value 3.10 times than that of CSE. However, to perform a more adequate economic evaluation of the cost of production of oil and phenolic-rich extracts of *M. flexuosa* using the extraction methods evaluated in the current study, as previously described [34], other factors such as raw material characteristics and seasonality, market size, product demand, and costs related to product quality control, packaging, and distribution must also be considered.

## 4. Conclusions

In the sensitivity study, the scenario with the greatest individual impact on the economic parameters was the reduction in the cost of raw materials by 50%. In this scenario, in the CSE and SFE+CSE, the COM decreased by 1.7 and 1.4 times, respectively. However, in both extraction processes, the COM reached its lowest value when scenarios 2 to 5 were merged, decreasing the COM value approximately two times in both extraction processes. Comparing both extraction processes, SFE+CSE was the most profitable economic process, because it made it possible to obtain two value-added by-products, oil and phenolic-rich extracts, with high nutraceutical value and desirable profit potential.

**Author Contributions:** Conceptualization, I.B. and L.O.-M.; methodology and investigation, I.B., L.O.-M., Z.C.-G., O.A.-C. and G.Z.; validation, I.B., L.O.-M. and G.Z.; writing—original draft preparation, I.B., Z.C.-G., O.A.-C. and G.Z.; writing—review and editing, I.B., L.O.-M. and G.Z.; supervision, I.B.; funding acquisition, I.B. All authors have read and agreed to the published version of the manuscript.

**Funding:** This research was funded by the National Fund for Scientific, Technological Development and Technological Innovation (FONDECYT) of the National Council of Science, Technology and Technological Innovation (CONCYTEC) of Peru, Contract 007-2018-FONDECYT-BM, and Universidad San Ignacio de Loyola.

**Data Availability Statement:** The data presented in this study are available on request from the corresponding author.

**Acknowledgments:** We thank the staff of the ICAN-USIL of the Universidad San Ignacio de Loyola for their valuable assistance in obtaining and processing the *M. flexuosa* samples.

**Conflicts of Interest:** The authors declare no conflict of interest.

## Nomenclature

| | |
|---|---|
| CAGR | Compound annual growth rate |
| CE | Catechin equivalents |
| COL | Cost of operational labor |
| COM | Cost of manufacturing |
| COU | Cost of utilities |
| CRM | Cost of raw material |
| CSE | Conventional solvent extraction |
| CUT | Cost of utilities |
| FCI | Fixed capital investment |
| FSE | Fluid supercritical extraction |
| GAE | Gallic acid equivalents |
| GEY | Global extraction yield |
| GM | Gross margin |
| IRR | Internal rate of return after taxes |
| NPV | Net present value at 7% |
| PBT | Payback time |
| ROI | Return on investment |
| S/F | Mass ratio of solvent to feed |

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
