# Peer review of "Production of Oil and Phenolic-Rich Extracts from Mauritia flexuosa L.f. Using Sequential Supercritical and Conventional Solvent Extraction: Experimental and Economic Evaluation"

_processes, doi:10.3390/pr10030459_

Round 1

Reviewer 1 Report

Overview and general recommendation:

I found the paper to be overall well written with interesting detailed economic evaluation which can be used for scale-up. However, there are some unclear data regarding optimized process parameters in SFE (pressure 200 bar, extraction temperature 42°C and CO2 flow rate 42 g CO2/min). In Table 2. it is stated that temperature used for oil extraction is 80°C and CO2 flow rate is 35 g CO2/min. Please elucidate. Also, abbrevation S/F is not defined, in Table 1. it represents solid/liquid ratio while in Table 2. it represents CO2 flow rate.  

Considering the fact that I detected inappropriate self-citations by authors, using their previously published proceeding paper, I recommend that a major revision is justified. 

Author Response

Point 1: I found the paper to be overall well written with interesting detailed economic evaluation which can be used for scale-up. However, there are some unclear data regarding optimized process parameters in SFE (pressure 200 bar, extraction temperature 42°C and CO2 flow rate 42 g CO2/min). In Table 2. it is stated that temperature used for oil extraction is 80°C and CO2 flow rate is 35 g CO2/min. Please elucidate. Also, abbrevation S/F is not defined, in Table 1. it represents solid/liquid ratio while in Table 2. it represents CO2 flow rate.

Response 1: Regarding the CO2 flow rate, the changes were made in Table 2. The definition of S/F was added in Table 1. Also, in Table 2, the S/F was replaced by the CO2 flow rate.

Point 2: Considering the fact that I detected inappropriate self-citations by authors, using their previously published proceeding paper, I recommend that a major revision is justified. 

Response 2: In line 13, the citation related to the proceeding paper was added on the first page of the article.

Reviewer 2 Report

This paper report a study about the production of extract from Mauritia flexuosa using 2 processes and focuses on the economic evaluation. I must say that it looks more an internal report of an engineering company than a research paper even if I do think that it is a good thing when research papers consider the extrapolation and economic evaluation. In this case the configuration is very specific because it is concluded that the raw material is 96% of the Cost of Manufacturing or the XXX. The cost of labor and the cost of utilities, in both cases is really negligible (0.69% and 0.51%) and the difference lies mainly in the YYY which is not the best configuration to compare the processes on an engineering point of view (it is on a business point of view). In consequence, I would relject this paper.

That said, the work is of good quality and, depending on the editorial criteria of the journal, it could be published. In this case, there are some awkwardness in the text that must be corrected.

1-line 63, Define supercritical adsorption. I checked the quoted paper (from Brunner et al.). In the present paper, it is said to be conventionally used but I do not think so, much less than supercritical extraction.

2- line 90. The raw material is lyophilized (and the extract too). This is very uncommon because lyophilisation is very energy consuming (more than five times a conventional drying). I know that finally it will be shown that energy is not a key factor, but lyophilisation remains a complex operation with a low productivity. Is there a specific reason (quality…) to use it?

3- In table 2, the S/F line does not give a solvent ratio (like in table 1), it should be kg CO2 used for 1 kg of processed material. It is a key factor to assess the energy consumption

4- Figure 3 and 4 giving the Gannt chart are useless here

5- In table 3, some line give a cost with fractions of dollar which is not reasonable in comparison to other lines with 2 significant figures.

6- In table 4, the wage cost is around 5dollars/hr. It is rather low compared to European ones (4 times higher with the social charges). May be worth mentioning if it could be critical.

7- In table 4 is given the price of ethanol and CO2. Ethanol recycling was mentioned line 310, 50% (rather low). What was the recycling rate for the CO2 ?

8- I was a bit surprised by the big difference of the CSE cost, 28 Mdollar, and the CSE+SFE (180 -28) Mdollar. Anti deflagrant equipment (use of ethanol) are expensive too.

9- When reading this paper on a black and white print (which is still the case for many people), the figure 4 and 5 are not understandable

10- Not all the financial abbreviations are known for all readers. Some of them are given under Table 6 and 7 but some are missing. Give their meaning as a list at the end of the paper.

Author Response

Point 1: 1-line 63, Define supercritical adsorption. I checked the quoted paper (from Brunner et al.). In the present paper, it is said to be conventionally used but I do not think so, much less than supercritical extraction.

Response 1: In line 63, the changes were made as suggested.

Point 2: line 90. The raw material is lyophilized (and the extract too). This is very uncommon because lyophilisation is very energy consuming (more than five times a conventional drying). I know that finally it will be shown that energy is not a key factor, but lyophilisation remains a complex operation with a low productivity. Is there a specific reason (quality…) to use it?

Response 2: Lyophilization was used to preserve the integrity and shelf life of the bioactive compounds of M. flexuosa, as well as to reduce the risk of contamination of the samples or extracts.

Point 3: In table 2, the S/F line does not give a solvent ratio (like in table 1), it should be kg CO2 used for 1 kg of processed material. It is a key factor to assess the energy consumption

Response 3: The S/F line was replaced by the CO2 flow rate as suggested.

Point 4: Figure 3 and 4 giving the Gannt chart are useless here

Response 4: Figures 3 and 4 were removed as suggested.

Point 5: In table 3, some line give a cost with fractions of dollar which is not reasonable in comparison to other lines with 2 significant figures.

Response 5: In Table 3, changes were made as suggested.

Point 6: In table 4, the wage cost is around 5dollars/hr. It is rather low compared to European ones (4 times higher with the social charges). May be worth mentioning if it could be critical.

Response 6: The average wage cost of factory workers in Latin America is US$ 5/h. In Table 4, to justify the cost of the local wage, reference 28 was added.

Point 7: In table 4 is given the price of ethanol and CO2. Ethanol recycling was mentioned line 310, 50% (rather low). What was the recycling rate for the CO2?

Response 7: The CO2 recycling rate was 90%.

Point 8: I was a bit surprised by the big difference of the CSE cost, 28 Mdollar, and the CSE+SFE (180 -28) Mdollar. Anti deflagrant equipment (use of ethanol) are expensive too.

Response 8: The difference between both extraction methods is mainly due to the SFE equipment used in the SFE+CSE process. The cost of this equipment was calculated based on the capacity of the SFE equipment and inflation, using the formula mentioned in reference 27. For the extraction volume of 2000 L, evaluated in this study, an equipment with a capacity of 600 kg/h was considered.

Point 9: When reading this paper on a black and white print (which is still the case for many people), the figure 4 and 5 are not understandable.

Response 9: Since Figures 2 and 3 were removed, considering these changes, the figures corresponding to COM, productivity and total capital investment were now 2 and 3. Changes were made to both figures as suggested.

Point 10: Not all the financial abbreviations are known for all readers. Some of them are given under Table 6 and 7 but some are missing. Give their meaning as a list at the end of the paper.

Response 10: The nomenclature was added at the end of the article.

Round 2

Reviewer 2 Report

All my previous remarks have been addressed in the present manuscript.

The paper can be published now.

This manuscript is a resubmission of an earlier submission. The following is a list of the peer review reports and author responses from that submission.

Round 1

Reviewer 1 Report

Dear Editor, Dear Authors,

I have carefully read the manuscript and find it very insightful and significant for the scientific community. I found that the topic is very interesting, the paper is straightforward, well-written, and well structured, the available scientific data are summarized and discussed clearly. In general, the paper clearly presents the research done.

Reviewer 2 Report

Work focused on the comparison of two extraction  methods of an active substance. A conventional extraction with a liquid solvent, and a supercritical extraction followed by a conventional extraction. Only two experiments are performed at non-optimized operating conditions. Subsequently, an economic analysis is carried out using a conventional simulation software. My proposal is to reject the work based on the following:
1. I believe that the manuscript is not appropriate for the topics of the journal Foods. Although the raw material is a food by-product, the objective of the paper is to comparatively analyze two extraction processes to obtain an active extract by a software. Perhaps it is more related to other journals as Processes.
2. The simulation of the process is based on considering that the results obtained at laboratory scale, are the same as those that would be obtained at industrial scale (line 135-138). There is no process scaling study that allows calculating the yield that would be obtained at industrial scale. If this consideration does not make sense, the simulation result is not adequate and only makes sense to compare both processes. 
3. Finally, after the whole simulation process, the conclusions is "Comparing both extraction processes, SFE+CSE was the most profitable economic process because it allowed obtaining two value-added by-products such as oil and phenolic-rich extracts with a high nutraceutical value and desirable profit". It is not necessary to conduct the study to conclude this. 
4. An economic analysis should include a study of the cost of the product in the market. Using SFE makes it possible to obtain a family of compounds different from those of liquid extraction with polar solvents. The purity of the product can change all the value of the product obtained. The market value of these products is reported to be between  3 and 5000 US$/kg. The high cost of the supercritical process could be justified if these products have a high market value. 
In general, the work focuses on analyzing the influence of each parameter on the final COM with sensitivity analyses of each of the variables. An optimization of the process, including economic aspects would be more interesting. 

Reviewer 3 Report

The purpose of the manuscript ID 1532926 was to produce oil and fruit extract of Mauritia flexuosa with the highest production efficiency and economy. However, some minor comments and considerations should be taken:

  • Page 2, lines 75, 75; Please change the sentence of "Therefore, the aim of this study was to compare the extraction yield and composition", because in this work, only the content of polyphenols and flavonoids in the tested extract was assessed. The composition assessment can be carried out with more advanced chromatographic techniques such as UPLC, HPLC, but these methods were not used in the work.
  • To optimize the extraction conditions, the Authors are asked to employ the Response Surface Methodology (RSM) method.
  • The Authors presented in the abstract, that "Pulp and oil are extracted from its fruits, with a high content of phenolic compounds and unsaturated fatty acids, respectively " (page 1, lines 13, 14). Please indicate the method used for the determination of unsaturated fatty acids and the place of presentation of these results.